# Quantification of DNA of *Fusarium culmorum* and Trichothecene Genotypes 3ADON and NIV in the Grain of Winter Wheat

**DOI:** 10.3390/pathogens11121449

**Published:** 2022-11-30

**Authors:** Tomasz Góral, Jarosław Przetakiewicz, Piotr Ochodzki, Barbara Wiewióra, Halina Wiśniewska

**Affiliations:** 1Plant Breeding and Acclimatization Institute—National Research Institute, Radzików, 05-870 Błonie, Poland; 2Institute of Plant Genetics, Polish Academy of Sciences, 34 Strzeszyńska Str., 60-479 Poznań, Poland

**Keywords:** chemotype, 3ADON, DNA, *Fusarium*, Fusarium head blight, NIV, real-time PCR, wheat

## Abstract

Fusarium head blight (FHB) is a wheat disease caused by fungi of the genus *Fusarium*. The aim of the study was to find relationships between the weather conditions in the experimental years and the locations and the amount of *F. culmorum* DNA and trichothecene genotypes, as well as the proportions between them. A three-year field experiment (2017, 2018 and 2019) was established in two locations (Poznań, Radzików). The DNA of *F. culmorum* was detected in all grain samples in an average amount of 20,124 pg per 1 μg of wheat DNA. The average amount of DNA from the 3ADON genotype was 4879 pg/μg and the amount of DNA from the NIV genotype was 3330 pg/μg. Weather conditions strongly affected the amount of DNA of *F. culmorum* and trichothecene genotypes detected in the grain. In the three experimental years, a high variability was observed in the coefficients of correlation between DNA concentrations and the FHB index, FDK, ergosterol and the corresponding toxins. There were significant correlations between disease incidence, fungal biomass (quantified as the total amount of fungal DNA or DNA trichothecene genotypes) and toxins (DON, 3AcDON and NIV) concentrations. The 3ADON trichothecene genotype dominated over the NIV genotype (ratio 1.5); however, this varied greatly depending on environmental conditions.

## 1. Introduction

Fusarium head blight (FHB) is a wheat disease (*Triticum aestivum* L.) caused by fungi of the *Fusarium* genus. These fungi infect wheat heads, causing necrosis of spikelets, infection and damage of the kernels, and contamination of tissues and grains with *Fusarium* toxins [1]. Several species associated with FHB have been identified, but the main species that infect wheat worldwide are *F. graminearum* Schwabe and *F. culmorum* (W.G. Sm.) Sacc. [2]. These species are highly pathogenic and can cause severe epidemics of FHB. The other species are medium or weakly pathogenic; however, they can also cause contamination of wheat grains with mycotoxins [3,4,5,6]. Phylogenetic studies of the collection of strains of *F. graminearum* have revealed that *F. graminearum* is a species complex consisting of at least 16 distinct species [7,8,9,10]. The most common species worldwide in the *F. graminearum* species complex (FGSC) is *F. graminearum sensu stricto*. In Europe *F. graminearum s.s.* was isolated almost exclusively [7,11]. Both species (*F. culmorum* and *F. graminearum* s.s.) belong to a large *F. sambucinum* species complex (FSAMSC) [12].

*Fusarium* species produce many toxins of different chemical groups. As small grain cereal contaminants, the most important are type A and type B trichothecenes (mainly deoxynivalenol = DON, nivalenol = NIV, T-2/HT-2 toxins) and the estrogenic compound zearalenone [13,14]. Within the group of type B trichothecene-producing species, three trichothecene genotypes (chemotypes) have been identified [15]. The NIV chemotype strains can produce NIV and its acetylated derivative. DON chemotype strains can produce DON and its acetylated derivatives (3-acetyl DON and 15-acetyl DON). Two subchemotypes are present within the DON chemotype: 3ADON strains produce DON and 3AcDON and 15ADON strains produce DON and 15AcDON. All three chemotypes have been detected in *F. graminearum* s.s. [10]. In *F. culmorum,* only 3ADON and NIV strains have been identified [16,17,18,19].

The NIV chemotype is generally considered less aggressive than the DON (3ADON, 15ADON) chemotypes [20,21,22,23,24]. However, the detailed results do not show such clear differences [21]. The pathogenicity of FGSC isolates was found to not depend on the type of toxin produced (DON versus NIV). Their aggressiveness was influenced mainly by the amount of toxins produced. The author stated that this is a key determining factor of the aggressiveness of isolates on wheat. Similar conclusions can be found in the article by Qu et al. [25] on the pathogenicity of FGSC isolates. In a paper published by Maier et al. [26] the progress of FHB in wheat heads was slower and symptoms were less severe for the NIV-producing isolate compared to the DON-producing isolates. However, both toxins were crucial for head infection, as isolates (DON and NIV) with disrupted *Tri5* gene did not show the spread of symptoms beyond inoculated spikelets. In field conditions Mesterhazy et al. [27] compared the aggressiveness of the NIV isolate with wheat using a set of isolates producing DON of *F. culmorum* and found no differences in the severity of symptoms of FHB and total trichothecenes production. Carter et al. [28] found no differences in the pathogenicity to wheat from DON- and NIV-producing isolates of FGSC (identified later as *F. asiaticum*). Interestingly, the NIV isolates were more pathogenic than the DON isolates to maize. 

DON and NIV are both aggressiveness factors and cause similar symptoms in wheat heads [29,30,31]. Despite this fact, differences have been found in the detoxification mechanism of both toxins. Lemmens et al. [29] suggested that different genes in the *Fhb1* gene cluster may be involved in resistance to these toxins. NIV is less toxic to wheat plants than DON and NIV isolates appear to be less aggressive [32,33]. In contrast, NIV is more toxic to humans and animals [34]. The amounts of NIV detected in agricultural products are lower than the amounts of DON, but both mycotoxins can co-occur and pose a threat to consumers when their total amount is above the maximum limits [35]. 

Resistance to FHB is a complex trait. Several types (mechanisms) of resistance have been identified. In their review article, Foroud et al. [36] described five types of resistance to FHB with two classes for type V. They are as follows. Type I—resistance to initial infection; Type II—resistance to FHB spread in the spike; Type III—resistance to kernel damage (infection); Type IV—tolerance to FHB or trichothecene toxins; Type V—resistance to accumulation of trichothecene toxins subdivided into: Class 1—by chemical modification (degradation or detoxification); Class 2—by inhibiting trichothecene synthesis. 

The evaluation of resistance type III is to determine the proportion of *Fusarium*-damaged kernels in the grain sample. This is done by dividing the sample into factions: kernels with signs of *Fusarium* damage (shriveled, white-discolored, pink, orange, carmine) and healthy-looking kernels [37]. This type of resistance can also be assessed by determining the content of ergosterol in the grain, which is a component of the cell membranes of fungi [38]. Its quantity indicates the amount of mycelium in the grain, which indirectly determines the degree of infection by *Fusarium* fungi. The amount of mycelium in the kernels can also be specified by measuring the concentration of *Fusarium* DNA in the grain using quantitative PCR (real-time PCR) [6,22,39]. This method is more precise because it can specifically detect DNA from *Fusarium* fungi (or selected *Fusarium* species, or chemotypes) in the grain [33,40,41]. The amount of *Fusarium* DNA can be used as a predictor of mycotoxin concentration in grains. However, it is strongly dependent on the material analyzed. No relationship has been found for naturally infected samples with low-toxin content [42]. In samples from inoculated heads or samples containing a large amount of toxins, these relationships have been found to be much stronger [43,44].

This research is based on the plant material described by us in the paper of Ochodzki et al. [45]. We selected 12 lines and cultivars that differ in resistance to FHB and subjected them to real-time PCR analysis for the amount of *F. culmorum* DNA, as well as to quantify the trichothecene genotypes 3ADON and NIV. The purpose of the investigation was to find relationships between the weather conditions in the experimental years, the locations, the amount of *F. culmorum* DNA and trichothecene genotypes, as well as the proportions between them. DNA concentrations were also related to resistance parameters of FHB and mycotoxin concentrations shown in our previous article [45].

## 2. Materials and Methods

### 2.1. FHB Inoculation Experiment and Mycotoxin Analysis

The plant materials included 12 winter wheat lines and cultivars: Winter wheat cultivars: Artist, Patras, and RGT Kilimanjaro.Breeding lines of wheat susceptible to FHB: KBP 14 16, NAD 10079, and SMH 8816Breeding lines/cultivars of wheat resistant to FHB: Fregata, NAD 13014, and NAD 13017Lines of wheat resistant to FHB that carry the *Fhb1* resistance gene: UNG 136.6.1.1 and S32.

A three-year field experiment (2017, 2018 and 2019) was established in two locations (Poznań, Radzików). At full anthesis (65 BBCH scale), the wheat lines were inoculated by spraying the heads with a spore suspension. Approximately two weeks after inoculation (depending on the appearance of symptoms of FHB) and one week later, the progress of the disease was visually evaluated using the FHB index (FHBi). At harvest, 20 randomly selected heads from each plot at each location were collected and threshed with a laboratory thresher. The percentage of *Fusarium*-damaged kernels (FDK) was visually scored. The weight of the FDK relative to the weight of the entire sample was marked as FDKw, and the number of FDK relative to the total sample size was marked as FDK#. 

Wheat grain samples from three inoculated plots were mixed and finely ground. The content of type B trichothecenes in the grain (DON, 3-acetyldeoxynivalenol (3AcDON), 15-acetyldeoxynivalenol (15AcDON) and NIV) was analyzed using the gas chromatography technique. Ergosterol (ERG) was chromatographically analyzed via high-performance liquid chromatography (HPLC) on a silica column using methanol. 

The details of the plant material, the field experiment methodology and the chemical analysis methodology were described in a previously published article [45].

### 2.2. DNA Quantification

In infected wheat grain samples (72 samples) from experiments in years 2017–2019, the amount of *F. culmorum* DNA and DNA from trichothecene genotypes 3ADON and NIV was analyzed. The real-time PCR technique was used. 

Quantitative analysis of plant DNA (a starter specific to the gene encoding the translation elongation factor *EF1α*) and fungal DNA (a starter specific to *F. culmorum*) was performed. (Table 1) [46]. Quantitative DNA analysis of two *F. culmorum* chemotypes (3ADON and NIV) was also performed. Primers specific to the *TRI12* gene belonging to the trichothecene biosynthesis gene cluster were applied [40]. 

Standard curves for *F. culmorum* species and chemotypes were determined using DNA obtained from pure fungal culture and isolates belonging to both chemotypes. Two *Fusarium culmorum* isolates were applied. KF846 was the 3ADON chemotype and KF350 was the NIV chemotype [43]. The isolates were grown on potato dextrose agar (PDA) medium covered with sterile polyethylene circles. The PDA plates were incubated at 22 °C with a 12 h photoperiod for one week. Pure mycelium was scraped from the polyethylene surface with a spatula.

#### 2.2.1. DNA Extraction

The materials for DNA extraction (mycelium or powdered grains) were ground with two stainless steel beads (5 mm) using a TissueLyser LT mill (Qiagen, Wrocław, Poland). Samples (100 mg) were used for DNA extraction, using QIAcube® automatic sample preparation for QIAGEN® spin column kits. DNA was extracted using the Plant Tissue Mini Protocol from the dNeasy Plant Mini Kit (Qiagen, Wrocław, Poland) and eluted in 100 µL of AE buffer.

The concentration of DNA from fungal isolates used for the standard curves was determined using a NanoReady Micro UV–Vis Spectrophotometer (Life Real, Hangzhou, China). The same method was used.

#### 2.2.2. Real-Time PCR

Real-time PCR was carried out in a total of 10.0 μL consisting of 2.0 μL 5× QUANTUM EvaGreen® HRM Kiter Mix (Syngen Biotech, Wrocław, Poland) and 2.0 μL template DNA. PCR reactions were performed in duplicate on all samples. Genomic DNA from grain samples was diluted 1:10 and pure cultures 1:100 before PCR.

All activities related to the preparation of the PCR reaction, dilution of DNA samples, and the preparation of standard curves were performed using the Myra Liquid Handling System (Bio Molecular System, Upper Coomera, Australia).

PCR was performed on a Mic qPCR (Bio Molecular System, Upper Coomera, Australia) using the following cycling protocol: 95 °C 15 min; 40 cycles of 95 °C for 15 s and 62 °C for 1 min followed by analysis of dissociation curves at 60 to 95 °C. For the plant assay, annealing and extension were carried out at 60 °C.

#### 2.2.3. Quantification of *Fusarium* DNA in Plant Material

Field samples were analyzed using specific assays together with the plant assay as a positive control. A standard curve was run for each of the assays with pure fungal DNA (Figure 1 and Figure 2). Five-fold dilution series of the isolates were used for standard curves. The same five-fold series of dilutions of plant DNA was used as a standard curve for the plant assay. The amount of fungal DNA was calculated from the values of the cycle threshold (Cq) using the standard curve, and these values were normalized with the estimated amount of plant DNA based on the plant *EF1α* assay.

The relative DNA content of *F. culmorum* and the chemotypes 3ADON and NIV was calculated in relation to wheat DNA (pg/μg).

### 2.3. Statistical Analysis

Statistical analysis was performed using XLSTAT Life Science, Version 2021.2.1.1119 (Addinsoft, New York, NY, USA). 

The concentrations of *F. culmorum* DNA and the 3ADON and NIV chemotypes were analyzed by variance analysis using the XLSTAT procedure: ANOVA. The variables did not follow a normal distribution according to normality tests. Data were transformed using the Box–Cox transformation. Three-way analysis of variance (ANOVA) was performed (year × location × line). Mean differences were determined according to Fisher’s LSD test at α = 0.05. Six experimental environments (year/location) were compared using the Kruskal–Wallis non-parametric test and the Steel–Dwass–Critchlow–Fligner multiple pairwise comparison method (XLSTAT procedure: comparison of k samples (Kruskal–Wallis, Friedman, …)). Relationships between *F. culmorum* DNA and 3ADON and NIV chemotypes, as well as FHBi, FDK, ERG, and mycotoxin concentrations were investigated using Pearson’s correlation tests (XLSTAT procedure: correlation tests).

## 3. Results

Weather conditions were highly variable in the experimental years and in the locations (Figure 3 and Appendix A). Most of all, it concerned rainfall distribution during the three-month coverage period from wheat heading to full maturity. The summary precipitation was the highest in 2017 in both locations. In Poznań it was similar in the following years; however, the distribution of rainfall was different. Rainfall was very low in June 2019, which was a period of wheat flowering and *Fusarium* inoculations. Only rainfall recorded was about a week after inoculation. In 2018 some rainfall was recorded during wheat flowering, but the most precipitation occurred after 20 June. In Radzików summary precipitation was lowest in 2019. However, rainfall in June was higher than in 2018 and occurred primarily during wheat flowering. In 2018, rainfall was recorded mostly after 20 June and was very low in one day during the wheat flowering period. 

The average temperature was the lowest in 2017 in both locations. It was higher in the subsequent years and similar in locations. In 2019 we observed very high temperatures in June after flowering and inoculation of the wheat heads.

*Fusarium culmorum* DNA was detected in all grain samples in an average amount of 20,124 pg per 1 μg of wheat DNA. The range was from 97,015 pg/μg (′SMH 8816′, Poznań, 2017) to 110 pg/μg (′RGT Kilimanjaro′, Radzików, 2018). The highest amount of *F. culmorum* DNA was detected in 2017 (41,484 pg/μg). It was much lower in the following years, 5682 pg/μg in 2018 and 1325 pg/μg in 2019. The means for the three years differed significantly. In two experimental locations, the amount of DNA in the Poznań grain was 24,862 pg/μg and from Radzików 15,385 pg/μg. The location means differed significantly. 

In the six experimental environments (year/location), the amount of *F. culmorum* DNA was the highest in Poznań in 2017. It did not differ significantly from the amount in Radzików in this year and differed significantly from the DNA amounts in the four other environments (Figure 4).

The wheat lines differed significantly in the amount of *F. culmorum* DNA in the grain (Table 2). The lowest concentration was found in the grain of two lines that carried the *Fhb1* resistance gene ′S 32′ and ′UNG 136.6.1.1′, as well as in the grain of the cultivar ‘Fregata’ and the breeding line ′NAD 13017′. The highest concentration of *F. culmorum* DNA was detected in the grain of the susceptible lines ′DL 358/3/4′ and ′KBP 1416′.

Regarding the trichothecene genotypes, the average amount of DNA from the 3ADON genotype was 4879 pg/μg in a range of 30,514 pg/μg (′DL 358/1/34′ Radzików, 2017) to 0 (′S 32′, Radzików, 2019). The highest amount of 3ADON DNA was detected in 2017 (9923 pg/μg). It was much lower in the following years: 2237 pg/μg in 2018 and 2532 pg/μg in 2019. The mean for 2017 differed significantly from the means for 2018 and 2019. In the two experimental locations, the amount of 3ADON DNA in the Poznań grain was 4846 pg/μg and in Radzików 4949 pg/μg. Location means did not differ significantly. In the six experimental environments (year/location), the amount of 3ADON DNA was the highest in 2017 in both locations (difference of means not significant) (Figure 5). It was lower in 2018 in Radzików and 2019 in Poznań, but not significantly different from the value for Radzików in 2017. The amount of 3ADON DNA was very low in 2018 in Poznań and in 2019 in Radzików and significantly lower than in the other environments.

The amount of DNA of the NIV genotype was 3330 pg/μg in a range of 27,131 pg/μg (′DL 358/1/34′ Poznań, 2017) to 8 pg/μg (′SMH 8816′, Poznań, 2018). The highest amount of NIV DNA was detected in 2017 (7131 pg/μg). It was much lower in the following years—1876 pg/μg in 2018 and 985 pg/μg in 2019. The mean for 2017 differed significantly from the mean for 2018 and 2019. In the two experimental locations, the amount of NIV DNA in Poznań grain was 5628 pg/μg and in Radzików it was five times lower (1033 pg/μg). The location means differed significantly. In the six experimental environments (year/location), the amount of NIV DNA was the highest in 2017 in Poznań (Figure 5). It was significantly lower in other environments. 

The lowest concentration of 3ADON DNA was detected in grain of two lines with the *Fhb1* gene. It was also low in two lines ′NAD 13017′ and ′NAD 13014′ and in the cultivars ‘Fregata’ and ‘RGT Kilimanjaro’ (Table 2). The highest concentration of 3ADON DNA was detected in the grain of the susceptible lines ′DL 358/3/4′ and ′KBP 1416′. The lowest concentration of NIV DNA was detected in the grain of line ′S 32′ with the *Fhb1* gene. It was low in the lines ′UNG 136.6.1.1′ (with *Fhb1*), ′NAD 13017′ and ‘Fregata’ cultivar. Similarly, for 3ADON DNA, the highest concentration of NIV DNA was detected in the grain of the susceptible lines ′DL 358/3/4′ and ′KBP 1416′.

The ratio of 3ADON DNA to NIV DNA was on average 1.5; however, it varied widely depending on year, location, and genotype. The lowest was 0.001 (‘RGT Kilimanjaro’ and ′S 32′, Radzików, 2019) and the highest 93.4 (′NAD 13017′, Radzików, 2017). In the three years, the ratio was as follows: 2017—1.4, 2018—2.3, and 2019—1.3. It was five times higher in Radzików (4.8) than in Poznań (0.9). In detail, the ratio in 2017 was 30 times higher in Radzików (31.8) than in Poznań (0.7). In 2018, it was 20 times higher in Radzików (5.6) than in Poznań (0.4). Only in 2019 was the ratio higher in Poznań (2.4) compared to Radzików (0.5). For wheat lines, the 3ADON/NIV DNA ratio was in the range of 0.8 (′UNG 136.6.1.1′)–2.6 (‘Fregata’). We found no relationship between line FHB resistance (head and kernels) and the 3ADON/NIV DNA ratio (Ochodzki et al., 2021). For example, for lines with the *Fhb1* gene, it was 1.7 for ′S 32′ and 0.8 for ′UNG 136.6.1.1′.

Next, we calculated the ratio of DNA (*F. culmorum*, chemotypes) to mycotoxin content in grain (data presented by Ochodzki et al. [45]). The ratio of *F. culmorum* DNA to type B trichothecenes was on average 2.4. It ranged from 0.1 (′NAD 13017′, 2018, Poznań) to 51.6 (′S 32′, 2018, Radzików). In the three years, the ratio was as follows: 2017—2.6, 2018—1.5, and 2019—3.2. It was similar in the two locations—2.2 in Radzików and 2.8 in Poznań. In detail, the ratio in 2017 was twice as high in Poznań (3.7) than in Radzików (1.7). In 2018, it was similar in the two locations (Radzików 1.7, Poznań 1.3). In 2019, the ratio was three times higher in Radzików (6.5) than in Poznań (2.0). There was a relationship of this ratio with phenotypic resistance to FHB [45]. For FHBi, FDKw and FDK# the coefficients were 0.527, 0.559, and 0560, respectively. They were not statistically significant. The ratio was the lowest for the two lines that carried the *Fhb1* resistance gene and the low-infected line ′NAD 13017′ (Table 2).

The 3ADON DNA to sum of DON toxins (DON, 3AcDON) ratio was 1.0. It varied from 0.002 (′S 32′, 2019, Radzików) to 62.9 (′S 32′, 2018, Radzików). In the three years, the ratio was as follows: 2017—1.0, 2018—1.4, and 2019—0.7. It was higher in Poznań (1.3) than in Radzików (0.8). In detail, the ratio in 2017 was twice as high in Poznań (1.8) than in Radzików (0.7). In 2018, it was similar in the two locations (Radzików 1.4, Poznań 1.0). In 2019, the ratio was lower and also similar in the locations (Radzików 0.7, Poznań 0.8). There was no relationship of this ratio to phenotypic resistance to FHB [45]. 

The ratio of NIV DNA to the sum of NIV toxin was 1.2. It varied from 0.001 (′SMH 8816′, 2018, Poznań) to 9.1 (′SMH 8816′, 2019, Radzików). In the three years, the ratio was as follows: 2017—1.0, 2018—1.4, and 2019—0.7. It was higher in Poznań (1.3) than in Radzików (0.8). In detail, the ratio in 2017 was seven times higher in Poznań (1.5) than in Radzików (0.2). In 2018, the ratio was 40 times higher in Radzików (12.2) than in Poznan (0.3). In 2019, the ratio was also higher in Radzików (3.8) than in Poznań (2.0). There was no relationship of this ratio with phenotypic resistance to FHB; however, it was the lowest for the two lines carrying the *Fhb1* resistance gene and the low-infected line ′NAD 13017′ (Table 2).

The concentrations of *F. culmorum*, 3ADON, and NIV DNA were correlated with phenotypic resistance (FHBi, FDK) and the concentration of ergosterol (ERG) and mycotoxins (DON, 3AcDON, NIV) in grain (data from Ochodzki et al. [45]. The amount of DNA from *F. culmorum* was significantly correlated with all variables (Table 3). The highest was the correlation coefficient with FHBi and ERG. Regarding mycotoxins, the coefficients were higher for DON and 3AcDON and lower for NIV.

The correlation coefficients of the trichothecene genotypes (3ADON, NIV) were lower than those of the *F. culmorum* DNA. However, the correlation coefficients of 3ADON DNA with DON concentration and NIV DNA with NIV concentration were higher than for *F. culmorum* DNA. The coefficient of correlation of the sum of DNA from the 3ADON and NIV trichothecene genotypes with the sum of trichothecenes was higher than for the DNA of the *F. culmorum* or the DNA of trichothecene genotypes independently correlated with trichothecenes. 

Since the coefficients were low, we analyzed the data for particular years separately (Figure 6). The coefficients for 2017 and 2019 were high. The results for 2018 were significantly different from those obtained for the other years. The correlation coefficients of DNA amount with phenotypic resistance and mycotoxins were very low. Similarly, the correlation coefficients for phenotypic resistance with mycotoxins were low.

## 4. Discussion

The weather conditions strongly affected the development of FHB and the amount of DNA detected in the grain. The 2017 was the most favorable for FHB. In this year the amount of *F. culmorum* DNA and 3ADON DNA was the highest in both locations. The story with NIV DNA was different. It was the highest in Poznań in 2017 but in Radzików it was the highest in 2019. It is difficult to explain this, but we can see that the amount of 3ADON genotype DNA was very low in 2019 in Radzików (which was different from previous years). It seems that the weather conditions favored the development of the NIV genotype. We compared daily weather conditions in Radzików in 2018 and 2019. In 2018, no rainfall was recorded after inoculation (beginning of June), but the second half of June was rainy with a lower temperature. On the contrary, in 2019, we recorded some rainfall during and after inoculations, but the end of June was dry and extremely hot. The strong association between environmental conditions and *Fusarium* biomass detected in mature grain was reported in other articles. Hoheneder et al. [47] found that environmental conditions before heading and late after flowering (fourth week) had the highest influence on *F. culmorum* in barley. Before flowering, the most important factors were temperature combined with sufficient rain and relative air humidity. After flowering, the most important factors were the sum of temperature, the sum of precipitation, and the relative humidity of the air. Xu et al. [48] found an increase in the amount of fungal biomass (the total amount of fungal DNA) with an increase in the length of the wetness period and temperature. For *F. culmorum*, it was 36 h of wetness accompanied by a temperature of 25 °C. They used the NIV chemotype of *F. culmorum* and observed that NIV production increased with increasing temperature and duration of wetness. The DON chemotype was the *F. graminearum* species, and in this case the main factor for the increase in the biomass and DON production was leaf wetness. Birr et al. [49] studied the effect of weather conditions on the amount of DNA of four *Fusarium* species that infect wheat. They found that the amounts of DNA depended mainly on precipitation and relative humidity during wheat flowering. The highest positive correlations were found for *F. graminearum*. They were weaker for *F. culmorum*. We did not have exact weather data; however, it was evident that precipitation in June (the wheat flowering period) had the greatest effect on the amount of *F. culmorum* DNA. Birr et al. [49] did not observe a significant effect of temperature on *Fusarium* DNA. We can draw a similar conclusion by comparing 2018 and 2019. The amounts of DNA did not differ significantly between the two years and we did not observe a significant effect of the extremely high temperatures recorded in 2019. The lower amount of *F. culmorum* DNA and DNA of the chemotypes compared to 2017 was mainly due to lower precipitation in June. It should be noted that the application of mist irrigation in Poznań influenced the results and it is difficult to compare them with data from natural conditions.

In the three experimental years, we observed large variability in the correlation coefficients between *F. culmorum* DNA and phenotypic resistance (FHBi, FDK) and ergosterol (ERG) and mycotoxin (type B trichothecenes) concentrations. The year 2018 was distinguished, in which the correlation coefficients were low and insignificant. This could be caused by the distribution of precipitation during wheat flowering when cereals are the most susceptible to *Fusarium* infection [50]. Inoculations were carried out at the full flowering stage. In 2018, in this period (1 June in Poznań and 6 June in Radzików) rainfall was very low or did not occur. After inoculation until 23 June, only one rain event was detected at both locations. In the paper we published, we found a highly significant correlation between the amount of *F. culmorum* DNA and the FHB index, as well as *Fusarium*-damaged kernels [43]. The coefficients for the toxins were more variable, low for DON and high for NIV. In general, they were similar to those obtained in this study (three-year data). However, we did not observe differences between the correlation of DON and NIV with *F. culmorum* DNA. 

The amount of trichothecene genotypes was significantly correlated with the corresponding toxins. The correlations 3ADON DNA vs. NIV and NIV DNA vs. DON were lower. Schner et al. [44] analyzed 300 wheat grain samples differing in DON content. They quantified the amount of DNA of trichothecene-producing *Fusarium* species using primers targeting the trichodiene synthase gene *Tri5*. They found a very high correlation coefficient between DNA and DON (0.956). We compared it with our results for trichothecene genotypes, as we targeted *Tri12*, the other gene for the trichothecene gene cluster (*Tri5*-cluster) [51]. As Schnerr et al. [44] used raw (non-transformed) data, we compared their results with coefficients calculated on non-transformed results. We found similar high coefficients for the correlations of 3ADON DNA vs. DON r = 0.803, 3ADON DNA vs. 3AcDON r = 0.807 and NIV DNA vs. NIV r = 0.811. Leišová et al. [52] quantified the DNA from *F. culmorum* in wheat and barley and compared its content with the DON content. They found a very high correlation for barley between the Ct values and the DON content. However, these were results for only one year. For wheat, they found a more complicated relationship. The determination coefficient (r^2^) varied widely in four experimental years from 0.327 to 0.888. However, we must add that the authors omitted results for one cultivar, which results deviated greatly from the linear relationship. Edwards et al. [53] inoculated wheat with different species of *Fusarium.* They did not find a correlation between the severity of FHB and the concentration of DON in the grain, but there was a good correlation (r^2^ = 0.76) between the amount of *Tri5* DNA (trichodiene synthase gene) and DON present in the grain. Sarlin et al. [54] analyzed different samples of Finnish barley grains. They found a significant correlation (r = 0.808) between *Fusarium* DNA and DON level in barley samples artificially inoculated with *F. culmorum*, *F. graminearum,* and *F. poae*. They applied primers that did not target genes involved in trichothecene production.

Under conditions of natural *Fusarium* infection, the prediction of the concentration of mycotoxin from *Fusarium* biomass amount in the grain is more problematic [55]. Analyzing wheat samples from four countries, Xu et al. [42] found a weak relationship between the amount of biomass of *F. graminearum* and *F. culmorum* and the concentration of mycotoxins. However, the authors claimed that the linear relationship with DON was nearly significant. In our study, we found a significant relationship between the amount of biomass of *F. graminearum* and the concentration of DON (r = 0.534) and zearalenone (r = 0.672) in wheat grain [56]. Analyzing Canadian wheat samples naturally infected with *F. graminearum* s.s., Demeke et al. [57] found a strong positive correlation between *F. graminearum* DNA and DON amount. Furthermore, other researchers observed highly significant correlations between the amount of *F. graminearum* DNA and the concentration of DON in naturally infected wheat grains [58,59]. Mentioned above, Sarlin et al. [54] found a low correlation (r =0.492) between DNA from trichothecene-producing fungi and DON in naturally infected Finnish barley grain. However, they found a high correlation (r = 0.967) for naturally infected barley samples from the USA and Canada. Yli-Matilla et al. [6] also analyzed cereal samples (barley, oats, wheat) from Finland. They found a high correlation between *F. graminearum* DNA and DON concentration in oats, but lower for wheat and barley. The authors explain that in Finland, *F. culmorum* is also an important DON producer in wheat and barley, but not in oats. Sarlin et al. [54] also showed that *F. graminearum* DNA was not correlated with the amount of DON in barley when this species was analyzed separately. Studying Italian wheat grain and whole grain, flour, and bread, Terzi et al. [60] found a significant correlation between *Fusarium* DNA and DON content. The coefficients were high, with r = 0.99 for all samples. The authors applied primers designed on the *Tri5*-*Tri6* intergenic sequence involved in DON synthesis. 

The ratio of the amount of DNA of the 3ADON chemotype to DNA from the NIV chemotype was on average 1.5. It varied widely depending on year, location, and genotype. For head inoculations, we applied a mixture of two 3ADON chemotype isolates and one isolate of NIV chemotype. Thus, without reduction of DNA of particular isolate due to competition ratio of 3ADON/NIV should be 2. The increase in the ratio was mainly due to the conditions in the experimental location in Poznań that were favorable for the NIV chemotype in two of the three experimental years. 

The reduction in the amount of a particular isolate in the mixed inoculation can be caused by competition between isolates. Xu et al. [48] inoculated wheat heads with *F. avenaceum*, *F. culmorum*, *F. graminearum*, and *F. poae*. They applied the species separately and in mixtures. Competition between species led to large reductions in fungal biomass compared to single-isolate inoculations. It was up to >90% reduction for the weaker species. On the contrary, mycotoxin production increased noticeably in co-inoculations, by as much as 1000 times. The authors stated that competition resulted in a higher production of trichothecene mycotoxins. They compared isolates of different species, but in the article by von der Ohe et al. [61] we can find comparison of two chemotypes of *F. graminearum,* DON, and NIV. The authors did not check whether the DON isolates belonged to the 3ADON or 15ADON sub-chemotype. In a mixed inoculation with DON and NIV isolates, the FHB rating was not significantly different from the binary DON+DON mixtures. Regarding mycotoxin production, the application of the DON+NIV isolate mixture resulted in the lack of DON in one environment and the production of DON similar to that for the DON+DON isolate mixtures in the other two environments. The authors did not analyze the NIV content, so we do not know what the effect on the mixed inoculation on amount of this toxin was. The explanation can be the result of the re-isolation from inoculated heads. In the first environment (no DON detected), the NIV chemotype dominated and in the others, the DON chemotype. We observed dominance of NIV (measured by DNA amount) in three out of the six environments (in Poznań in 2017 and 2018 and in Radzików in 2019). This was strongly dependent on weather and experimental conditions (mist irrigation).

Miedaner et al. [62] compared aggressiveness and mycotoxins production of two pairs of *F. culmorum* isolates of the 3ADON and NIV chemotype. On average, they observed that the aggressiveness of the isolate mixtures was significantly lower than that of the isolates applied individually for inoculation. Similarly, mycotoxin concentrations were significantly lower in the mixtures in most of the comparisons. The authors re-isolated and molecular identified *F. culmorum* isolates from inoculated heads. They found the dominance of the highly aggressive DON isolate over the NIV isolate and the less aggressive DON isolate. However, the more aggressive NIV isolate dominated both the less aggressive NIV and the DON isolates. Unfortunately, the authors did not test mixtures of more aggressive isolates of both chemotypes and presented mycotoxin data only for one environment. Therefore, it is difficult to evaluate the effect of experimental conditions on the frequency of chemotypes and proportions of mycotoxins. 

## 5. Conclusions

The DNA of *F. culmorum* and trichothecene chemotypes (3ADON, NIV) was quantified in winter wheat. A strong effect of the weather conditions in three experimental years on the amounts of DNA was observed. The amount of DNA was significantly correlated with parameters of phenotypic resistance to FHB and with the concentration of trichothecene toxins. The coefficients were very high in the two experimental years. In one year, the correlation coefficients of the amount of DNA with phenotypic resistance and mycotoxins were very low, showing a strong effect of weather conditions on these relationships. Coefficients were higher for DNA from *F. culmorum.* However, for mycotoxins (DON, NIV, total trichothecenes), the coefficients were higher for DNA from trichothecene chemotypes. The content of DNA of 3ADON chemotype was on average 1.5 times higher than the NIV chemotype. It varied widely depending on environmental conditions, and in some cases the amount of DNA of NIV chemotype was higher than that of the 3ADON chemotype. 

## Figures and Tables

**Figure 1 pathogens-11-01449-f001:**
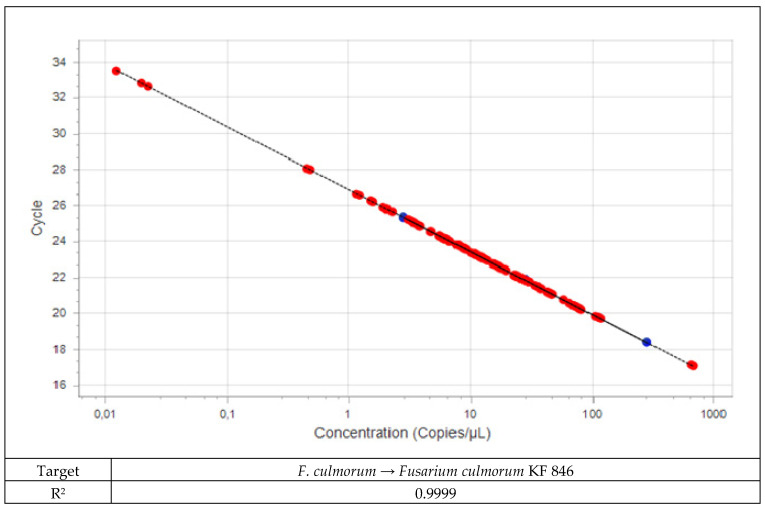
Standard curve for DNA from *Fusarium culmorum*.

**Figure 2 pathogens-11-01449-f002:**
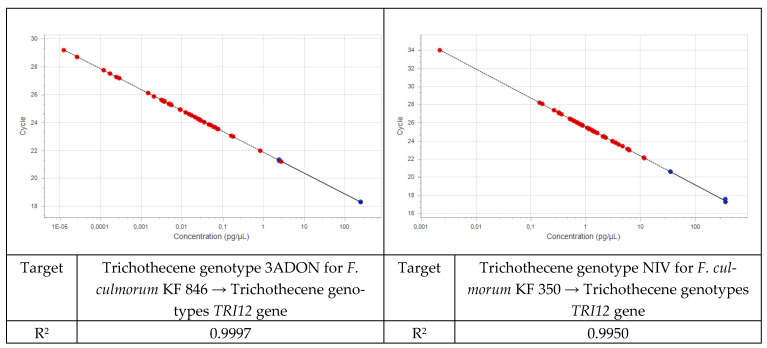
Standard curves for DNA of the 3ADON and NIV trichothecene genotypes.

**Figure 3 pathogens-11-01449-f003:**
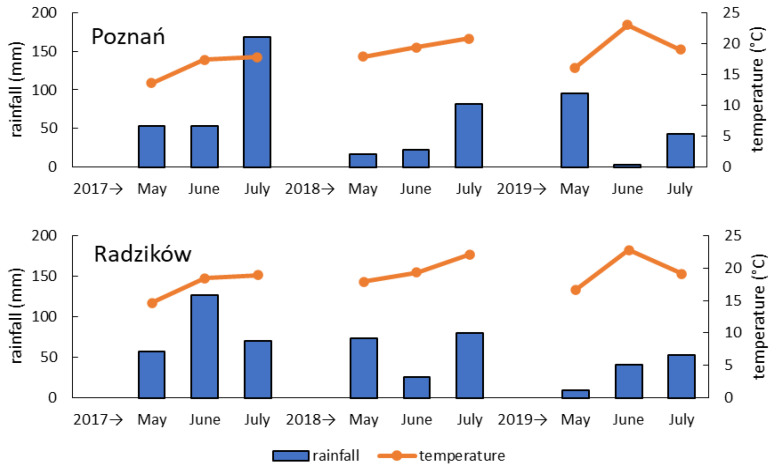
Monthly rainfall and average temperature in May, June and July 2017–2019 in the two experimental locations (Poznań, Radzików).

**Figure 4 pathogens-11-01449-f004:**
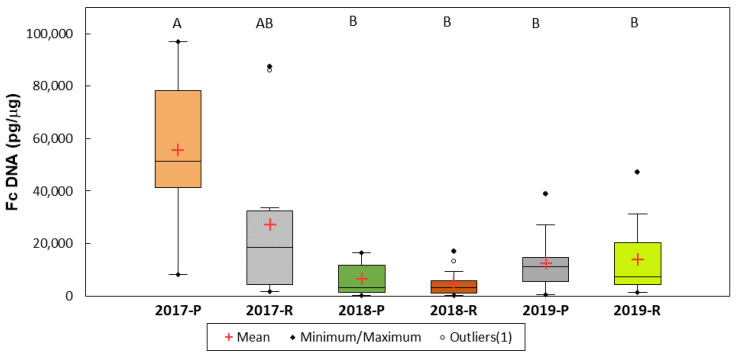
Concentration of *F. culmorum* DNA in wheat grain in six experimental environments (year location). The environment means marked with the same letter are not significantly different at *p* = 0.05 according to the Kruskal–Wallis test and the multiple pairwise comparison method of Steel–Dwass–Critchlow–Fligner.

**Figure 5 pathogens-11-01449-f005:**
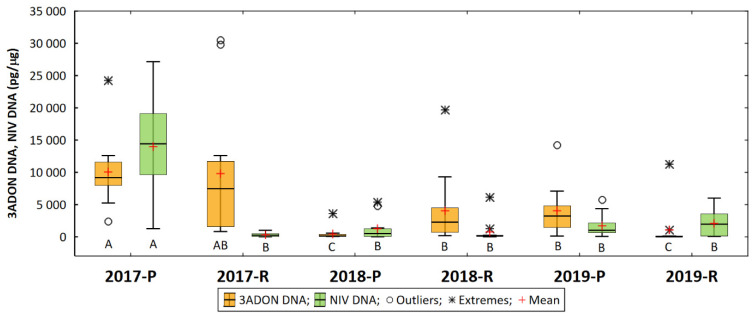
DNA concentration of two trichothecene genotypes of *F. culmorum* (3ADON, NIV) in wheat grain in six experimental environments (year/location). Means marked with the same letter are not significantly different at *p* = 0.05 according to the Kruskal–Wallis test and the Steel–Dwass–Critchlow–Fligner multiple pairwise comparison method.

**Figure 6 pathogens-11-01449-f006:**
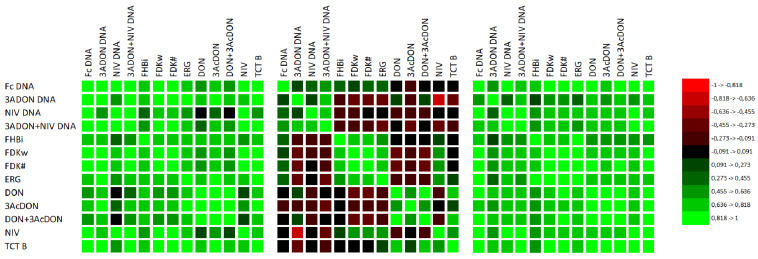
Matrixes of coefficients of correlation of DNA concentrations of *F. culmorum* and trichothecene genotypes (3ADON, NIV) with phenotypic resistance (FHBi, FDK) and ergosterol (ERG) and mycotoxin (type B trichothecenes) concentrations in three experimental years (2017, 2018, and 2019) [45]. The critical value for Pearson’s correlation coefficient at *p* = 0.05 is equal to r = 0.404.

**Table 1 pathogens-11-01449-t001:** Primers used for the real-time PCR amplification of the *F. culmorum,* 3ADON, and NIV chemotypes in the *TRI12* gene and the translation elongation factor gene 1α (*EF1α*).

Target	Primer Name	Sequence (5′-3′)
*F. culmorum*	Fcul F	CACCGTCATTGGTATGTTGTCACT
	Fcul R	CGGGAGCGTCTGATAGTCG
3ADON	3ADONf	AACATGATCGGTGAGGTATCGA
	3ADONr	CCATGGCGCTGGGAGTT
NIV	NIVf	GCCCATATTCGCGACAATGT
	NIVr	GGCGAACTGATGAGTAACAAAACC
Plant EF1α	Hor1F	TCTCTGGGTTTGAGGGTGAC
	Hor2R	GGCCCTTGTACCAGTCAAGGT

**Table 2 pathogens-11-01449-t002:** Amount of DNA (pg fungal DNA/μg plant DNA) of *F. culmorum* and 3ADON and NIV chemotypes in wheat grains and ratios of the amounts of *F.c,* 3ADON, and NIV DNA to concentrations of type B trichothecenes, DON and NIV.

Line	*F. culmorum* DNA (pg/μg)	3ADON Chemotype DNA (pg/μg)	NIV Chemotype DNA (pg/μg)	*F.c.* DNA/ TCT B ^#^	3ADON DNA/ DON ^#$^	NIV DNA/ NIV ^#^
S 32 (*Fhb1*)	2763 a	1104 a	662 a	1.1	1.5	0.4
Fregata	8118 abc	3094 a	1189 a	2.4	1.1	1.9
UNG 136.6.1.1 (*Fhb1*)	9438 ab	1727 a	2041 a	1.9	1.0	0.6
NAD 13017	9465 abc	2437 a	1641 a	1.6	0.8	0.6
NAD 13014	14,625 bcd	2720 a	2961 a	3.1	1.0	1.6
RGT Kilimanjaro	17,674 bcd	3068 a	3208 a	3.3	0.9	1.7
Artist	21,309 cde	4207 a	3362 a	2.2	0.6	1.3
NAD 10079	21,872 cde	7768 a	4574 a	2.7	1.4	1.7
Patras	24,355 cde	5010 a	3701 a	2.0	0.6	1.0
SMH 8816	26,996 cde	6841 a	4476 a	3.1	1.3	1.3
DL 358/13/4	41,502 de	11,236 a	6130 a	2.4	0.9	1.1
KBP 1416	43,369 e	9556 a	6019 a	3.5	1.2	1.4
Mean	20,124	4897	3330	2.4	1.0	1.2

^#^ mycotoxin data from Ochodzki et al. [45]; ^$^ sum of DON and 3AcDON; means marked with the same letter are not significantly different at *p* = 0.05 according to Fisher’s LSD test performed on transformed variables.

**Table 3 pathogens-11-01449-t003:** Coefficients of correlation between DNA concentrations of *F. culmorum* and trichothecene genotypes (3ADON, NIV) with phenotypic resistance (FHBi, FDK) and ergosterol (ERG) and mycotoxin (type B trichothecenes) concentrations.

Variables (n = 72)	Fc DNA	3A DON DNA	NIV DNA	3A DON+NIV DNA	FHBi ^$^	FDKw ^$^	FDK# ^$^	ERG ^$^	DON ^$^	3Ac DON ^$^	DON+ 3Ac DON ^$^	NIV ^$^
3ADON DNA	0.533											
NIV DNA	0.707	0.370										
3ADON+NIV DNA	0.767	0.801	0.769									
FHBi	0.602	0.268 *	0.323 **	0.413								
FDKw	0.513	0.384	0.369	0.455	0.655							
FDK#	0.538	0.376	0.404	0.454	0.677	0.985						
ERG	0.619	0.436	0.352	0.477	0.716	0.813	0.829					
DON	0.531	0.552	0.295 *	0.552	0.381	0.108ns	0.112 ns	0.344 **				
3AcDON	0.609	0.459	0.400	0.549	0.426	0.209 ns	0.216 ns	0.396	0.790			
DON+ 3AcDON	0.534	0.549	0.300 **	0.554	0.380	0.108 ns	0.111 ns	0.345 **	1.000	0.798		
NIV	0.497	0.277 *	0.514	0.460	0.411	0.625	0.665	0.669	0.368	0.494	0.373	
TCT B	0.519	0.488	0.422	0.541	0.417	0.427	0.452	0.608	0.806	0.618	0.808	0.755

The coefficients are significant at *p* < 0.001 except for those marked with * (*p* < 0.05), ** (*p* < 0.01) or ns (non-significant); TCT B—sum of DON, 3AcDON and NIV; ^$^ data from Ochodzki et al. [45].

## Data Availability

Not applicable.

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
