# Peer review of "Quantification of DNA of Fusarium culmorum and Trichothecene Genotypes 3ADON and NIV in the Grain of Winter Wheat"

_pathogens, 2022, doi:10.3390/pathogens11121449_

Round 1
Reviewer 1 Report
Fusarium head blight (FHB) is a wheat disease caused by fungi of the Fusarium genus, which produce many trichothecene toxins, such as NIV, DON, 3/15-ADON, and so on. In this study, 12 lines and cultivars that differ in resistance to FHB from years 2017-2019 and two locations were subjected to real-time PCR analysis for the amount of F. culmorum DNA, as well as to quantify the trichothecene genotypes 3ADON and NIV. It is obvious that the temperature and rainfall were different from 2017 to 2019 in the two locations. However, the detailed relationships between the weather conditions ( temperature and rainfall) and the amount of F. culmorum DNA and trichothecene genotypes should be shown in Abstract and Result.
In addition, there is some details need to revise:
1. The desciption of significant differences and figures in L223-224、L246-247、L262-263 were incorrect, which should be checked.
2.“The was significantly of DNA amount”in L466 should be rewritten.
3. References cited in recent 5 years are less (12/58). The author should cite more new literature to increase the value of this article.
Reviewer 2 Report
The article deals with the Fusarium head blight and the impact of weather changes on the presence of the fungi and the toxins- trichothecenes they produce in wheat grains. Research is relevant to agricultural production and economy as well as to public health, as trichotecenes are dangerous to human and animals. The topic is therefore very important and falls within the scope of Pathogens.
The authors pursued their goal by field experiment – inoculation of different winter wheat lines and cultivars (FHI-resistant and not) with Fusarium spores, and thereafter visual examination of FHB development as well as testing of grain samples for type B trichothecenes and ergosterol (chromatography technics) as well as genomic DNA and DNA encoding trichothecene toxins by real-time PCR. The research was conducted over three years in two different localities, so material collected for analysis is relatively wide. The experimental part – material, methods and results are described properly, the only mistake was in statistical methods (Kruskal-Wallis, Friedman, …). The figures and tables are comprehensible and clear. The authors found significant correlations between disease incidence, fungal biomass and toxins concentrations, as well as indicated that the 3ADON trichothecene genotype dominates over the NIV genotype and that variation in toxins amount greatly depending on environmental conditions. The conclusions are in line with the results obtained, and the introduction and discussion are extensive and comprehensive, referring to a rich literature (58 items).
The only improvement I can suggest , is to prepare a graphic summary, which in the case of such a work rich in experiments, could be helpful to readers.
